# Limb–Girdle Muscular Dystrophies Classification and Therapies

**DOI:** 10.3390/jcm12144769

**Published:** 2023-07-19

**Authors:** Camille Bouchard, Jacques P. Tremblay

**Affiliations:** 1Departement de Médecine Moléculaire, Université Laval, Quebec, QC G1V 0A6, Canada; camille.bouchard@crchudequebec.ulaval.ca; 2Centre de Recherche du Centre Hospitalier Universitaire de Quebec, Quebec, QC G1E 6W2, Canada

**Keywords:** limb–girdle muscular dystrophy, LGMD, classification, therapy

## Abstract

Limb–girdle muscular dystrophies (LGMDs) are caused by mutations in multiple genes. This review article presents 39 genes associated with LGMDs. Some forms are inherited in a dominant fashion, while for others this occurs recessively. The classification of LGMDs has evolved through time. Lately, to be considered an LGMD, the mutation has to cause a predominant proximal muscle weakness and must be found in two or more unrelated families. This article also presents therapies for LGMDs, examining both available treatments and those in development. For now, only symptomatic treatments are available for patients. The goal is now to solve the problem at the root of LGMDs instead of treating each symptom individually. In the last decade, multiple other potential treatments were developed and studied, such as stem-cell transplantation, exon skipping, gene delivery, RNAi, and gene editing.

## 1. Introduction

Limb–girdle muscular dystrophies (LGMD) are muscular dystrophies that affect skeletal muscles, mostly proximal (hips and shoulder muscles). They are caused by a mutation in a gene encoding a protein, which is specific to each subtype. LGMD inheritance is autosomal. This is in contrast to Duchenne or Becker muscular dystrophies, which are caused by a mutation in the DMD gene located on the X chromosome [1]. Some LGMD forms are dominant, and others are recessive.

At the beginning of the 1800s, cases of muscular atrophy were reported to medical authorities. At the end of the 19th century, a correlation was established whereby several members of families developed comparable muscular degeneration [1]. In 1876, Leyden chose to classify muscular dystrophies into two categories; the first one affecting children, described by Duchenne, and the other one causing a progressive degeneration in adults. Then, a facioscapulohumeral category and a distal category were added to the classification in the early 1900s.

In the 1950s, Walton enumerated 18 types of myopathic disorders:(1)Pseudohypertrophic muscular dystrophy (Duchenne; Gowers);(2)The pelvic girdle atrophic type (Leyden–Mobius);(3)The juvenile (scapulohumeral) type (Erb);(4)The facioscapulohumeral type (Landouzy–Dejerine);(5)The distal type (Gowers);(6)The late juvenile type (Nevin);(7)The Barnes type;(8)Myotonia congenita (Thomsen) and paramyotonia congenita (Eulenburg);(9)Dystrophia myotonica (Steinert; Batten and Gibb);(10)The simple atrophic type of amyotonia congenita (Batten)—the congenital myopathy of Aldren Turner;(11)The local myopathies (Bramwell; Denny-Brown; Weaver and Maun);(12)Ocular myopathy (Hutchinson; Fuchs; Kiloh and Nevin);(13)Menopausal muscular dystrophy (Shy and McEachern);(14)Benign childhood myopathy (probably polymyositis) (Walton and Nattrass);(15)Unusual myopathies of metabolic origin (McArdle; Acheson and McAlpine);(16)Thyrotoxic myopathy;(17)“Myasthenic myopathy”;(18)Atypical myopathy;

In 1953, Stevenson categorized the pathology as ‘autosomal limb–girdle muscular dystrophy’ and characterized the condition among 27 families in Ireland [2].

In the 1990s, subtypes of the condition were established by finding which protein was deficient and locating the responsible mutation in the gene [3]. The subtypes were then classified by a number. This indicates whether the inheritance is autosomal dominant (LGMD 1) or recessive (LGMD 2). A letter follows to specify the mutated gene. Theses names are given in **bold** in the following text.

The new classification suggested by Straub et al. (2018) is LGMD. This is followed by either R or D to indicate recessive or dominant inheritance, as well as by a number determined by the order of discovery. The names are underlined in the following text.

Mitsuhashi et al. reviewed the latest discoveries on LGMD types in 2012 [4] and further research was performed during the following decade. Taghizadeh et al. presented a more recent table in their review in 2019 [5]. We also included the new classification of LGMDs proposed by Straub et al. (2018) [6]. This classification considers new LGMDs, such as Bethlem myopathy, due to their ability to respect the LGMD criteria. However, it also reclassifies pathologies which were previously considered as LGMD into another category for not respecting all criteria. The definition of an LGMD is currently the following:

“Limb girdle muscular dystrophy is a genetically inherited condition that primarily affects skeletal muscle leading to progressive, predominantly proximal muscle weakness at presentation caused by a loss of muscle fibres. To be considered a form of limb girdle muscular dystrophy the condition must be described in at least two unrelated families with affects individuals achieving independent walking, must have an elevated serum creatine kinase activity, must demonstrate degenerative changes on muscle imaging over the course of the disease, and have dystrophic changes on muscle histology, ultimately leading to end-stage pathology for the most affected muscles” [6].

## 2. Classification of LGMDs

### 2.1. Autosomal Dominant

For example, **LGMD 1A** is characterized by a mutation in the MYOT gene, responsible for myotilin synthesis [7,8] which cross-links actin filaments and controls sarcomere assembly [9]. However, this disease is not included in the new classification by Straub et al. (2018) because the weakness is distal and does not affect proximal muscles [6]. In fact, the most frequent phenotype is a late-onset distal myopathy affecting ankles, feet or calves [10]. Cases of respiratory insufficiency or cardiac failure are also reported [11].

**LGMD 1B** is caused by mutations in the LMNA gene coding for Lamin A/C [12]. This in turn causes alterations in the nuclear envelope in fibroblasts [13]. Straub et al. (2018) do not consider this myopathy as an LGMD because it is associated with cardiac arrhythmia [6]. It is therefore not included in the new nomenclature since the main effect is not exerted on proximal muscles.

**LGMD 1C** occurs due to mutations in the CAV3 gene [14], causing a caveolin-3 deficiency associated with skeletal muscle weakness due to impaired mitochondrial form and function [15]. This pathology was excluded from the new classification due to the muscle rippling and myalgia [6]. A recent Korean study showed that the phenotype is variable, being mild for some patients and more severe for others. HyperCKemia is found in all patients, whereas myopathic features are less frequent and can occur as ankle contracture, calf hypertrophy, exercise intolerance, or muscular cramps [16].

**LGMD 1D** (now LGMD D1) is characterized by a mutation in the DNAJB6 gene, which encodes for a co-chaperone protein implicated in sarcomeric protein maintenance and aggregation [17]. Patient muscle magnetic resonance imaging (MRI) shows fat infiltration, and biopsies show an increased number of internal nuclei and rimmed vacuoles [18]. The affected muscles are both proximal and distal; the lower and proximal limbs are affected most severely, while the effects are mild in the distal and upper ones.

**LGMD 1E** occurs due to mutations in the DES gene encoding for desmin [19]. A mutation in this protein can lead to desmin aggregate formation and/or to the development of an irregular muscle fibre shape [20]. LGMDs are not considered in the new classification because of distal weakness and cardiomyopathy [6]. Desminopathy shows fatty replacement in semitendinosus, gastrocnemius and soleus MRI. The typical phenotype is characterized by distal weakness, but facial and bulbar weakness were also reported [21]

**LGMD 1F** (now LGMD D2) is associated with mutations in the TNPO3 gene coding for transportin 3 [22]. This protein function is still being studied, but abnormalities in sarcomeric assembly have been observed via patient biopsies [23]. The age of onset and severity of phenotype have a high variability. Typically, however, a pelvic lower girdle weakness is noticed first, followed by the atrophy of the axial muscle and shoulder girdle. MRI also shows fatty replacement of fibres in pelvic and thigh muscles. Biopsy results show rimmed vacuoles and enlarged nuclei [24].

**LGMD 1G** (now LGMD D3) is caused by a defect in the heterogenous nuclear ribonucleoprotein D-like (hnRNPDL) protein [25] and patient muscles show rimmed vacuoles [26]. The protein has been shown to regulate transcription and alternative splicing [27].

**LGMD 1H** is another autosomal dominant type of LGMD with an alteration in a gene situated in chromosome 3 (3p23–p25). However, it remains unassociated with a specific protein [28] and therefore cannot be classified as an LGMD since all the characteristics one are not fulfilled [6].

**LGMD 1I** (now LGMD D4) is caused by dominant mutations in the CAPN3 gene, encoding for calpain 3. Research shows that missense mutations lead to a milder phenotype and null mutations to increased phenotype severity [29]. Fat infiltration int the affected muscles (thigh adductors and semimembranosus, but also calf gastrocnemius and soleus) is also noticed in MRI and happens before the deterioration of muscle function [30].

**Bethlem myopathy** (now LGMD D5) is now also classified as an LGMD and can be caused by a dominant mutation in COL6A1, COL6A2 or COL6A3 genes, encoding for the Collagen 6 protein that plays a role in extracellular matrix structure and muscle regeneration [6]. This myopathy also causes there to be a higher fat fraction in the muscles, especially in the psoas major [31], and recent authors have suggested that this be used as an MRI diagnosis tool. Other MRI patterns also indicate this pathology, such as a “target sign” located in the *rectus femoris* or a “sandwich sign” in the *vastus lateralis* [32].

### 2.2. Autosomal Recessive

As for autosomal recessive limb–girdle muscular dystrophies, **LGMD 2A** (now LGMD R1) is caused by recessive mutations in CAPN3, a gene encoding for calpain 3 [33], which could regulate the sarcomere [34]. The dominant form is also described above as LGMD 1I/D4.

**LGMD 2B** (LGMD R2) is associated with mutations in the DYSF gene that impairthe synthesis of dysferlin, a protein responsible for sarcomere stability and muscular repair [35]. The posterior thigh and leg muscles are the most affected areas and show a progressive fat fraction increase, which is accentuated when patients become non-ambulant [36]. Two-thirds of patients have a diamond-shaped bulge on their quadriceps when in action [37].

**LGMD 2C** (LGMD R5) is caused by mutations in the SGCG gene, which synthetizes the γ-Sarcoglycan protein. This protein is part of the DAP complex and plays a role in the cytoskeleton extracellular matrix link [4]. The phenotype severity varies in siblings with the same mutation and is not related to the age of onset [38]. The phonotype is considered severe when patients lose ambulation before 13 years of age.

**LGMD 2D** (LGMD R3) is also caused by a mutation in a gene of the sarcoglycan complex: the SGCA gene responsible for the synthesis of the α-Sarcoglycan protein. The main signs include pelvic girdle weakness, fatty replacement, and atrophy of the *vastus intermedius*, *semimembranosus* and *biceps femoris* muscles [39].

**LGMD 2E** (LGMD R4) is associated with a mutation in the SGCB gene, related to the β-Sarcoglycan protein. The phenotype is a weakness and fatty replacement in: *Latissimus dorsi*, spine extensors, abdominal belt, glutei, adductors, and various thigh muscles [40].

**LGMD 2F** (LGMD R6) is caused by a mutation in the SGCD gene coding for the δ-Sarcoglycan protein. The phenotype is variable, but generally involves a progressive atrophy of proximal muscles and high serum creatine kinase. Some patients also present cardiomyopathy [41].

**LGMD 2G** (LGMD R7) is related to mutations in the TCAP gene encoding for telethonin, which plays a role in myofibrillogenesis in interaction with titin in the sarcomeric Z-disk [42,43]. The first sign of this is generally thigh muscle weakness progressively also affecting calf muscles via fatty infiltration in the glutei, hip and thigh muscles [44].

**LGMD 2H** (LGMD R8) is known for mutations in the TRIM32 gene, which codes for the tripartite motif-containing protein 32 responsible for the movement of Ca^2+^ in skeletal muscles [45]. Patients present calf hypertrophy and weakness. Their muscles contain fibres of various size, small vacuoles, and increased numbers of internal nuclei [46].

**LGMD 2I** (LGMD R9) is caused by mutations in the FKRP gene, coding for fukutin-related protein. This protein is necessary for the glycosylation of α-dystroglycan [47]. Muscle biopsies show different fibre modifications: some are hypertrophic, while others are split, hyaline, rounded or necrotic [48].

**LGMD 2J** (LGMD R10) is caused by mutations in the TTN gene. This codes for titin, a protein responsible for muscle tension [49]. Patients have weakened thighs and their muscle biopsy shows rimmed vacuoles containing amyloid accumulation [50].

**LGMD 2K** (LGMD R11) and **LGMD 2N** (LGMD R14) are due to mutations in the POMT1 and POMT2 genes, respectively. These genes allow for the synthesis of protein O-mannosyltransferase 1 and 2, of which co-expression is necessary for O-mannosylation, an important reaction in α-dystroglycan [51]. LGMD 2J (LGMD R10) shows proximal limbs weakness and cognitive impairment [52].

**LGMD 2L** (LGMD R12) is associated with mutations in the ANO5 gene, which encodes for the Anoctamin 5 protein necessary for achieving functional calcium-activated chloride channels in the muscles [53]. The phenotype is one of prominent asymmetrical *quadriceps femoris* and *biceps brachii* atrophy [53].

**LGMD 2M** (LGMD R13) is linked to a mutation in the FKTN gene. The fukutin protein also plays a role in the glycosylation of α-dystroglycan [54]. This LGMD causes proximal muscle weakness in early childhood, with the occurrence of hypotonia and/or hypertrophy. Some patients have ocular problems, cognitive impairment or cardiomyopathy [55].

**LGMD 2N** (LGMD R14) is caused by a mutation in the POMT2 gene necessary for the synthesis of protein O-mannosyltransferase 2. The phenotype is generally responsible for walking difficulties and pain during exercise. Hamstrings, paraspinal and gluteal muscles are the first parts of the body affected, leading to a strength reduction in hips as well as knee flexors and extensors. Cognitive impairment is also reported [56].

**LGMD 2O** (LGMD R15) is caused by mutations in the POMGnT1 gene, which codes for protein O-mannose beta-1,2-Nacetylglucosaminyltranserase 1. Former **POMGNT2-related muscular dystrophy** (LGMD R24) was also called Walker–Warburg syndrome or muscle–eye–brain disease and is now categorized as an LGMD since it corresponds to all LGMD new criteria. It is caused by a mutation in the POMGnT2 gene, encoding for protein O-mannose beta-1,2-Nacetylglucosaminyltranserase 2. Both protein O-mannose beta-1,2-Nacetylglucosaminyltranserase 1 and 2 catalyse a modification of α-dystroglycan [57]. POMGnT1 is necessary for the synthesis of the M1 core glycan structure. MGAT5B is then added to form the M2 core, and the addition of POMGnT2 then transforms it into the M3 core [57]. **LGMD 2O** (LGMD R15) patients have proximal limb muscle weakness around puberty. The affected muscles are calves, quadriceps, hamstrings and deltoids. Myopia is also reported among patients [58].

**LGMD 2P** (LGMD R16) occurs due to mutations in the DAG1 gene, encoding for dystroglycan, a protein that is part of the dystroglycan complex [5]. In this case, a proximal muscle weakness is reported along with cognitive impairment [59].

**LGMD 2Q** (LGMD R17) occurs when there is a mutation in the PLEC1 gene. This gene encodes for the synthesis of the plectin protein. This has many roles, including cell survival, cell growth, actin organization and T-cell activation [5]. Progressive limb muscle weakness is reported in deltoids, *erector spinae*, glutei, *biceps femoris* and *adductor magnus*. Some plectinopathies include skin affectation, but case studies do not include skin problems [60].

**LGMD 2R** is not classified as an LGMD any longer, and this is caused by a mutation in the DES gene encoding for desmin. Its role is to connect sarcomeres together in order to form myofibrils [5]. It was rejected from the LGMD category since the weakness it induces is distal, not proximal [6].

**LGMD 2S** (LGMD R18) occurs due to mutations in TRAPPC11. This encodes for transport protein particle complex 11, which is responsible for the transport between endoplasmic reticulum and Golgi apparatus [5]. The onset occurs during school age and the proximal muscle weakness is progressive. The hips are more affected than shoulders and the development of a respiratory restrictive disorder is reported in some patients [61].

**LGMD 2T** (LGMD R19) is associated with the GMPPB gene, encoding for the GDP-mannose pyrophosphorylase B protein. Its function is to produce GDP-mannose for the O-mannosylation of α-dystroglycan [62]. Paraspinal and hamstring muscles are the most affected by this, as seen per MRI, and the phenotype can cause hypotonia, epilepsy, cognitive impairment, cataracts and cardiomyopathy [63].

**LGMD2U** (LGMDR20) is related to mutations in the ISPD gene. Its protein is isoprenoid synthase, responsible for glycosylation of α-dystroglycan [64]. The phenotype includes sural hypertrophy and moderate proximal weakness [65].

**LGMD2V** is associated with mutations in the GAA gene. This encodes for α-1,4-Glucosidase, an important protein for lysozyme function [5]. It is not categorized as an LGMD and is now called Pompe disease [66]. The phenotype varies from mild to severe, its effects include a progressive proximal muscle weakness, and it can also involve the respiratory or cardiac muscles [67].

**LGMD2W** is caused by mutations in the LIMS2 gene, which synthetizes Lim and senescent cell antigen-like domains 2 protein. This protein is involved in cell spreading and migration [5]. It was only reported in one family and, therefore, cannot be considered an LGMD yet. This is because Straub et al. (2018) consider it an LGMD only when 2 or more unrelated families have the same pathology [6]. The patients develop and early onset proximal weakness. This includes calf hypertrophy, leading to severe quadriparesis during adolescence. A dilated cardiomyopathy was also reported in both patients [68].

**LGMD2X** is related to the BVES gene for blood vessel endocardial substance. This encodes for Popeye domain-containing protein 1 (POPDC1) that is involved in the structure and function of cardiac and skeletal muscle cells [5]. This myopathy was recently reported in 5 more families, making it part of the LGMDs [6]. An article from May 2022 classifies **LGMD2X** as LGMDR25 (OMIM 616812) [69]. The reported symptoms are exercise-induced myalgia of the thighs and hips, as well as cardiac arrhythmia.

**LGMD2Y** is associated with mutations in the TOR1A1P1 gene, known for the Torsin 1A-interacting protein 1. This protein serves as a link between the nuclear membrane and lamina during cell division [5]. It is not considered an LGMD since it has been reported in only one family [6]. Its symptoms include skeletal muscle weakness and cardiac involvement [70].

**LGMD2Z** (LGMDR21) is caused by mutations in the POGLUT1 gene for protein O-glucosyltransferase 1, which is involved in protein transport [5]. The phenotype induces limb–girdle muscle weakness as well as a decrease in α-dystroglycan glycosylation and fatty replacement in muscles [71].

Three more myopathies are now considered as LGMDs:

**Bethlem myopathy recessive** (LGMDR22), caused by mutations in collagen 6 genes COL6A1, COL6A2, and COL6A3, which play roles in the extracellular matrix structure and muscle regeneration [6]. The phenotype is one of a progressive proximal weakness and ankle contracture. The subtle contracture of the interphalangeal joint is a specific sign of Bethlem myopathy [72].

**Laminin α2-related muscular dystrophy** (LGMDR23), associated with mutations in the LAMA2 gene, which is expressed as Laminin α2, a protein involved in myotube stability and apoptosis [73]. The phenotype is a progressive proximal muscle weakness and accounts for 36% of the patients reported having seizures without an epilepsy-related gene [74].

**POMGNT2-related muscular dystrophy** (LGMDR24) occurs due to mutations in the POMGnT2 gene, protein O-linked mannose β-1,4-N-Acetylglucosaminyl-transferase 2, which catalyzes reactions in α-dystroglycan [57]. The phenotype is one of a progressive proximal lower limb muscle weakness, along with possible ocular problems and cognitive impairment [75].

Table 1 shows the classification of LGMD types, comparing the mutated gene, its associated protein, and its function.

## 3. Therapies

### 3.1. Symptomatic Treatments

Limb–girdle muscular dystrophies (LGMDs) have different symptoms, even between members of a family with the same mutation. Treatments addressing specific symptoms are available in some cases, such as night ventilation for impaired respiratory function or β-blockers for cardiac symptoms [76,77]. Adapting nutrition to patients needs and physical rehabilitation are also used to address these issues [78].

### 3.2. Molecular and Gene Therapy

The last two decades were marked by rapid progress in the molecular and genetic therapies [79,80]. Table 2 summarizes the therapeutic techniques that can be considered as options for treating LGMDs.

#### 3.2.1. Stem-Cell Transplantation

Stem-cell transplantation consists in injecting cells containing the correct gene to express the deficient protein. There are two stem-cell transfer possibilities; autologous stem-cell transfer or allogenic stem-cell transfer [81]. Autologous stem-cell transfer consists in taking the patients cells and treating them in vitro to restore their specific protein expression and re-implant them in the patient. Allogenic stem-cell transfer is the injection of a healthy donor’s stem cells into the patient. Myoblast transplantation (or myoblast transfer) consists in injecting myoblasts from a healthy donor locally in the patient’s muscle to restore the deficient protein [82,83].

#### 3.2.2. Exon Skipping

Exon skipping uses antisense oligonucleotides (ASOs), which are small DNA sequences with different chemical properties. The first ASO generation had a phosphoribose backbone, which was rapidly degraded by endonucleases and exonucleases. The backbone was then modified to obtain second-generation ASOs, which are less prone to degradation due to the replacement of the non-binding hydrogen atoms with sulfur ones. This modification resulted in a longer half-life of ASOs in serum, which can be explained by their improved nuclease resistance and serum protein binding capacity. Some ASOs also include another modification in 2′ of the ribose molecule, such as 2′Omethyl (2′OMe) and 2′Omethoxy-ethyl (2′MOE), which hybridize more efficiently to the target RNA and modify its expression. Another group is used to modify splicing or to inhibit translation. This group does not contain the typical deoxyribose in the phosphoribose backbone, but a morpholino ring instead. In addition, the linkage is an uncharged phosphorodiamidate one instead of a charged phosphodiester one. These properties make the ASOs even more resistant to nuclease and protease degradation. The target region can be degraded by RNAse H (category I: RNAse H competent), or bind to block a start site, an RNA binding protein, a splicing site, or an upstream open reading frame (uORF) (category II: steric block) [84]. Category I uses an RNAse H1 enzyme. This recognizes the DNA (ASO)-RNA complex and cleaves the binding site to degrade the matched RNA. Approved ASOs from this category include fomivirsen, mipomersen and inotersen [85]. Category II uses a steric block, which strongly binds to the target and masks this sequence during splicing or translation. Approved ASOs from this category include eteplirsen, golodirsen and nusinersen.

ASOs can be delivered using different methods of pairing the ASO (1) to Triantennary N-acetylgalactosamine (GalNAc), (2) to peptides, (3) with lipids such as cholesterol, (4) with antibodies or adaptamers, (5) or to a stimuli-responsive structure. It is also possible to pack the ASO into (6) a stable nucleic acid lipid particle or intp an (7) exosome, (8) spherical nucleic acid nanoparticle made with a gold core linked to ASOs with metal–thiol, or (9) a DNA cage with an ASO at its end [85].

Clinical trials performed using ASOs show various degrees of efficacy. Scholars conducted a phase I/II clinical trial using AVI-4658, a morpholino ASO, to skip the exon 51 from dystrophin in DMD patients (age 10 to 15). The patients received different doses of AVI-4658 through nine injections in the extensor digitorum brevis (EDB), a foot muscle. A biopsy of this muscle was conducted 28 days later to determine if dystrophin, which had previously been absent, could then be synthetized in the muscle. Immunohistochemistry and Western blots confirmed the presence of dystrophin in the fibers around injection sites [86]. Another phase I/II trial with the same treatment injected intravenously in 5- to 15-year-old patients on a weekly basis for 12 weeks also showed up to 55% dystrophin-positive fibers for one patient, although the average differed [87]. For LGMDs, ASOs were proven to be effective at skipping exon 32 in dysferlin in vitro, but clinical trials have not been carried out on humans yet [88].

#### 3.2.3. Gene Delivery

One avenue in research consists of delivering the healthy gene to synthetize the deficient protein [78]. For example, Bartoli et al. (2006) delivered the calpain gene (CAPN3) to mice intramuscularly using a recombinant adeno-associated virus (rAAV) vector [89,90]. In 2010, Mendell et al. observed the sustained expression of the injected α-sarcoglycan gene when delivered with an AAV at 6 months post-treatment in voluntary patients in a phase I clinical trial entitled NCT00494195 [91]. In 2013, Bartoli et al. delivered the gene systemically with an AAV but suppressed its expression in the cardiac tissue to avoid cardiac toxicity [89,90]. Also, in 2013 Xu et al. (2013) restored α-dystroglycan glycosylation and phenotype in FRKP mutant mice using an rAAV9-FKRP [92]. The result was the improved muscular contractility of the mice gastrocnemius and restored strength of the soleus. In 2023, Seo et al. also systemically delivered the γ-sarcoglycan gene, using an AAV to restore the protein expression and function in mouse muscles [93]. A clinical trial began recently to evaluate the safety of SRP-9003 and its ability to restore the expression of β-SG in the skeletal muscles of LGMD2E/R4 patients (NCT05876780). SRP-9003 delivers the gene coding for the β-SG protein in an AAV (rAAVrh74.MHCK7.hSGCB).

However, some proteins, such as dysferlin, have a gene that is superior in size to the delivery capacity of AAV (about 4.7 kb). Therefore, a dual-AAV is required to deliver the dysferline gene (6.9 kb) [78,94]. Overlapping the two cDNAs for a 1 kb region seems to ease the reconstitution from the 2 AAVs [95]. The treatment resulted in dysferlin overexpression in the muscles and the restoration of the muscles’ ability to repair their membrane after injury [94].

Other teams tried to deliver the gene directly as plasmidic DNA without using an AAV vector. Guha et al. injected plasmidic DNA intramuscularly in mice, followed by an electroporation [96]. Their results were positive for restoring calpain, dysferlin and α-sarcoglycan. This method could be used to prevent an immune response from patients who were already in contact with many AAV serotypes.

#### 3.2.4. RNAi

In 2012, Wallace et al. used RNA interference (RNAi) to knock down the dominant mutation in myotilin (MYOT), causing LGMD1A [97]. This led to an undetectable amount of the mutant protein 3 months post-treatment in mice, as well as a gain in strength and muscle mass.

#### 3.2.5. Gene Editing

In the 1990s, double-strand breaks (DSBs) were introduced in genomes with I-SceI, a homing endonuclease that promotes homologous recombination (HR) [98].

Zinc-finger nucleases (ZFNs) create a specific DSB and induce HR or non-homologous end-joining (NHEJ) by folding a 30-amino acid sequence into a ββα structure stabilized by zinc [99]. A zinc-finger domain can bind to a DNA triplet by inserting the α-helix into the DNA double helix [99]. Many domains can cover a longer DNA sequence and form a set. Two sets form a dimer, which induces a DSB. A correct donor sequence can then replace the original sequence by HR [100].

Transcription activator-like effector nucleases (TALENs) are also used as a dimer to induce a DSB. A TALEN repeat domain recognizes only 1 nucleotide instead of 3, meaning that they can target any sequence [101]. They also cause less cytotoxicity than ZNFs [101].

Correcting the gene directly in the cells is also the aim of using clustered regularly interspersed short palindromic repeats (CRISPR) paired with a CRISPR-associated (Cas) nuclease [102]. The CRISPR–Cas system was initially identified in prokaryotes. These use an RNA sequence to identify the bacteriophage infecting them and degrade the pathogenic DNA [102]. The CRISPR technique is now used to identify a DNA or RNA sequence and create a specific double strand break. In fact, the targeted sequence matches the complementary region RNA (crRNA), which further anneals to a trans-activating CRISPR–RNA (tracrRNA). TracrRNA plays a role in the maturation of the CRISPR–RNA by pairing the preCRIPR-RNA, which will then be cleaved to become the guide for Cas9. Jennifer A. Doudna and Emmanuelle Charpentier fused both tracrRNA and crRNA together as a 18–24 nucleotide single-guide RNA (sgRNA), also including a Cas9 domain to cleave a custom target sequence [103,104]. The system could be used for different applications, such as therapy for hereditary diseases [105].

Base editing was developed to change all bases of the same type in the target window into other specific bases without cleaving. For example, all A bases would change for G bases in the specific window [106]. Base editors use a CRISPR–Cas system and a cytidine or an adenosine deaminase and target either RNA or DNA [107].

The prime editing technology was developed several years after base editing technology. It uses a Cas9 nickase fused with a reverse transcriptase and a prime editing guide RNA (pegRNA) to cleave one DNA strand and specifically modify a sequence without using a donor DNA [103,108]. The pegRNA includes a reverse transcriptase template (RTT) and a primer binding site (PBS). The complex binds to the DNA, and Cas9 nickase cuts one strand. Then, the PBS hybridizes to the cleaved strand, which allows the sequence to be completed by transcribing the RTT sequence. The 3′ or 5′ flap is then incorporated into the final DNA strand [103,108].

Some in vitro experiments allowed us to correct mutations causing LGMDs, such as LGMD2A (LGMDR1) [109] and LGMD2B (LGMDR2) [110].

## 4. Conclusions

Limb–girdle muscular dystrophies are caused by mutations in various genes. Dominant and recessive forms cause a predominant proximal muscle weakness that may be found in two or more unrelated families. In the last decade, different techniques have been investigated for use to treat LGMDs. For now, only symptomatic treatments are available for the patients. However, multiple other potential treatments are under investigation, such as stem-cell transplantation, exon skipping, gene delivery, RNAi, and gene editing.

## Figures and Tables

**Table 1 jcm-12-04769-t001:** Classification of LGMD types and affected genes, associated protein and function.

	Type	Gene	Protein	Function
Type 1: autosomal dominant	LGMD1A	MYOT	Myotilin	Cross-links actin filaments + sarcomere assembly
	LGMD1B	LMNA	Lamin A/C	Nuclear envelope in fibroblasts
	LGMD1C	CAV3	Caveolin 3	Skeletal muscle mitochondria form and function
	LGMD1D (LGMDD1)	DNAJB6	DNAJ homologue, family B, member 6	Sarcomeric protein maintenance and aggregation
	LGMD1E	DES	Desmin	Maintain muscular structure and function
	LGMD1F (LGMDD2)	TNPO3	Transportin 3	Sarcomeric assembly
	LGMD1G (LGMDD3)	HNRNPDL	Heterogenous nuclear ribonucleoprotein D-like	Regulate transcription and alternative splicing
	LGMD1H	?	?	?
	LGMD1I (LGMDD4)	CAPN3	Calpain 3	Sarcomere regulation
	Bethlem myopathy dominant (LGMDD5)	COL6A1, COL6A2, COL6A3	Collagen 6	Extracellular matrix structure and muscle regeneration
Type 2: autosomal recessive	LGMD2A (LGMDR1)	CAPN3	Calpain 3	Sarcomere regulation
	LGMD2B (LGMDR2)	DYSF	Dysferlin	Sarcomere stability and muscular repair
	LGMD2C (LGMDR5)	SGCG	γ-Sarcoglycan	Cytoskeleton extracellular matrix link
	LGMD2D (LGMDR3)	SGCA	α-Sarcoglycan	Cytoskeleton extracellular matrix link
	LGMD2E (LGMDR4)	SGCB	β-Sarcoglycan	Cytoskeleton extracellular matrix link
	LGMD2F (LGMDR6)	SGCD	δ-Sarcoglycan	Cytoskeleton extracellular matrix link
	LGMD2G (LGMDR7)	TCAP	Telethonin	Myofibrillogenesis
	LGMD2H (LGMDR8)	TRIM32	Tripartite motif-containing protein 32	Ca^2+^ movement of in skeletal muscles
	LGMD2I (LGMDR9)	FKRP	Fukutin-related protein	Glycosylation of α-dystroglycan
	LGMD2J (LGMDR10)	TTN	Titin	Muscle tension
	LGMD2K (LGMDR11)	POMT1	Protein O-mannosyltransferase 1	O-mannosylation in α-dystroglycan
	LGMD2L (LGMDR12)	ANO5	Anoctamin 5	Calcium-activated chloride channels
	LGMD2M (LGMDR13)	FKTN	Fukutin	Glycosylation of α-dystroglycan
	LGMD2N (LGMDR14)	POMT2	Protein O-mannosyltransferase 2	O-mannosylation in α-dystroglycan
	LGMD2O (LGMDR15)	POMGnT1	Protein O-mannose beta-1,2-Nacetylglucosaminyltranserase	Catalyzes reactions in α-dystroglycan
	LGMD2P (LGMDR16)	DAG1	Dystroglycan	Part of the dystroglycan complex
	LGMD2Q (LGMDR17)	PLEC1	Plectin	Cell survival, cell growth, actin organization and T-cell activation
	LGMD2R	DES	Desmin	Connect sarcomeres to form myofibrils
	LGMD2S (LGMDR18)	TRAPPC11	Transport protein particle complex 11	Transport between endoplasmic reticulum and Golgi apparatus
	LGMD2T (LGMDR19)	GMPPB	GDP-mannose pyrophosphorylase B	Production of GDP-mannose for the O-mannosylation of α-dystroglycan
	LGMD2U (LGMDR20)	ISPD	Isoprenoid synthase	Glycosylation of α-dystroglycan
	LGMD2V	GAA	α-1,4-Glucosidase	Lysozyme function
	LGMD2W	LIMS2	Lim and senescent cell antigen-like domains 2	Cell spreading and migration
	LGMD2X (LGMDR25)	BVES	Popeye domain containing protein 1 (POPDC1)	Structure and function of cardiac and skeletal muscle cells
	LGMD2Y	TOR1A1P1	Torsin 1A-interacting protein 1	Link between the nuclear membrane and lamina during cell division
	LGMD2Z (LGMDR21)	POGLUT1	Protein O-glucosyltransferase1	Protein transport and process.
	Bethlem myopathy recessive (LGMDR22)	COL6A1, COL6A2, COL6A3	Collagen 6	Extracellular matrix structure and muscle regeneration
	Laminin α2-related muscular dystrophy (LGMDR23)	LAMA2	Laminin α2	Myotubes stability and apoptosis
	POMGNT2-related muscular dystrophy (LGMDR24)	POMGnT2	Protein O-linked Mannose β-1,4-N-Acetylglucosaminyl-transferase 2	Catalyzes reactions in α-dystroglycan

**Table 2 jcm-12-04769-t002:** Molecular and gene therapies studied for LGMD.

Technique	Description
Stem-cell transplantation	Systemic or local injection of stem cells.
Exon skipping	Using antisense oligonucleotides to skip the mutated exon and produce a truncated functional protein.
Gene delivery	Delivery of a gene to the cell with local or systemic injection of a vector, e.g., an AAV.
RNAi	An interfering RNA knocks down the mRNA of the mutant allele.
Gene editing	Recognizing a specific DNA or RNA sequence and modifying it.

## Data Availability

Not applicable.

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
