# Peer review of "Limb–Girdle Muscular Dystrophies Classification and Therapies"

_jcm, 2023, doi:10.3390/jcm12144769_

Round 1
Reviewer 1 Report
The Authors reviewed LGMDs classifications and available therapies.
The aim of the paper is not clear and not stated in the introduction, nor in the abstract which is completely missing.
I would suggest some major revisions:
- include the abstract and the keywords in the text;
- Information provided from line 78 to 198 (Classification of LGMDs) almost recapitulate what is listed in Table 1. I would extend the text with some clinical information/case series/case reports/muscle MRI pattern about the different LGMDs to make the text more meaningful;
- The second section (Therapies) is missing some relevant information to the topic: i) I would include comments and citations of previous researches on genome-editing approach by CRISPR technique applied to in vitro LGMD models; ii) gene therapy has reached the clinical setting for some LGMDs, therefore ongoing clinical trials on sarcoglycanopathies (LGMDR3, LGMDR5) need to be included in this section (cite the clinical trial identifier in clinicaltrial.gov).
The quality of English Language needs extensive revisions, especially in the first part of the review (Introduction and Classification of LGMDs).
Author Response
Hello,
I followed all your suggestions allong with the other reviewer's ones. Here is the new version of my manuscript.
Thank you for taking the time to read my work and help me improve it.
I activated the revision mode on Microsoft Word so that you can see the modifications I made
-include abstract and keywords: done
-extend text with some clinical information/case series/case reports/muscle MRI pattern
I added a few sentences to each paragraph describing the MRI pattern or affected muscles as well as case reports on diagnosis
-Therapies: i) include information about in vitro LGMD researches : I added Some in vitro experiments allowed to correct mutations causing LGMDs such as LGMD2A (LGMDR1) {Stefanie, 2023, Cas9-induced single cut enables highly efficient and template-free repair of a muscular dystrophy causing founder mutation} and LGMD2B (LGMDR2) {Turan, 2016, Precise Correction of Disease Mutations in Induced Pluripotent Stem Cells Derived From Patients With Limb Girdle Muscular Dystrophy}.
- ii) include clinical trials for LGMD:
Clinical trials using ASOs show various efficacy. A Phase I/II clinical trial using AVI-4658, a morpholino ASO to skip the exon 51 from dystrophin in DMD patients (age 10 to 15). They received different doses of AVI-4658 through nine injections in the extensor digitorum brevis (EDB), a foot muscle. A biopsy of this muscle was made 28 days later to determinate if dystrophin, which was previously absent, could then be synthetized in the muscle. Immunohistochemistry and Western blots confirmed the presence of dystrophin in the fibres around injection sites [86]. Another Phase I/II trial with the same treatment injected intravenously in 5 to 15 year-old patients on a weekly basis for 12 weeks also showed up to 55% dystrophin-positive fibres for one patient, but the average [87]. For LGMDs, ASOs were proven effective to skip exon 32 in dysferlin in vitro, but clinical trials have not been done on humans yet [88].
A clinical trial began recently to evaluate the safety of SRP-9003 and its ability to restore the expression of β-SG in the skeletal muscles of LGMD2E/R4 patients (NCT05876780). SRP-9003 delivers the gene coding for the β-SG protein in an AAV (rAAVrh74.MHCK7.hSGCB).
In 2010, Mendell et al. observed a sustained expression of the injected α‐sarcoglycan gene delivered with an AAV 6 months post-treatment in voluntary patients in a Phase I clinical trial NCT00494195[91].

Reviewer 2 Report
This is a comprehensive paper on the new classification of the LGMD disease spectrum based on the ENMC guidelines from 2017. Nomenclature of disease is an important educational topic for both clinicians and patients. It has to be underlined that widely accepted nomenclature should only be changed with caution and by consulting key clinical experts and patient advocacy groups. Because there is a spectrum of phenotypes under the same genetic entity, and a wide genetic heterogeneity under the same phenotype, it is important to identify appropriate selection criteria to be used when diagnosing patients for the proteins and genes responsible for the subtype of LGMD.
Therefore, this overview gives a good tool for the clinicians to assess the new subtypes of LGMD with short genetic background, including the new discoveries and gives a useful matching to the former classification.
Critical comments:
- Abstract is missing.
- Line 81: “and to control sarcomere assembly” – controls
- Line 85: Is LGMD1B included in the new nomenclature or not? Needs explanation.
- Line 123: LGMD2D should be in a new line, also LGMD2E and LGMD2F for better overview in listing.
- Line 139: „which co-expression is” - of which
- Line 147: What was the original classification of POMGNT2-related muscular dystrophy?
- Line 150: „catalyze their own modification of the α-dystroglycan” - this is not clear. What is meant under own modification? The process of the protein O‐mannosylation should be shortly explained.
- Lines 220-224: AON therapeutic options should be discussed in somewhat more details, e.g. different chemistries, mode of deliveries, efficacies in clinical trials for the subtypes where developments are already available.
- Lines 225-242: In the gene therapy descriptions it is not clear whether the observations are made in mice experiments or in clinical trials. Most probably in mice but this has to be clearly stated in each description.
- Line 249: Again, mice or human application of RNAi approach?
- Line 272: The abbreviation tracrRNA has to be explained, see trans-activating crispr RNA. Some more explanation of the small trans-encoded RNA would be beneficial.
- In general, it would improve the paper to give e more comprehensive overview of the different therapeutic options with special focus on the LGMD subtypes where the those approaches are in progress.
Author Response
Hello,
I followed all your suggestions allong with the other reviewer's ones. Here is the new version of my manuscript.
Thank you for taking the time to read my work and help me improve it.
Abstract is missing: I added it
Line 81: controls
Line 85: I added: It is therefore not included in the new nomenclature since the main effect is not on proximal muscles.
Line 123: I separated in different lines
Line 139 of which
Line 147: POMGNT2-related muscular dystrophy was also called Walker-Warburg Syndrome Or Muscle-Eye-Brain Disease.
Line 150 : explain the process : Both Protein O-mannose beta-1,2-Nacetylglucosaminyltranserase 1 and 2 catalyse a modification of the α-dystroglycan [57]. POMGnT1 is necessary for the synthesis of the M1 core glycan structure. MGAT5B is then added to form the M2 core and POMGnT2 then transforms it into the M3 core.
Lines 220-224 (now 313+) : add details on AON
Exon skipping uses antisense oligonucleotides (ASOs), which are small DNA se-quences with different chemical properties. The first ASO generation has a phosphoribose backbone, which was rapidly degraded by endo-nucleases and exonucleases. The back-bone was then modified to obtain second-generation ASOs, which are less prone to deg-radation due to the replacement of the non-binding hydrogen atoms with sulfur ones. This modification results in a longer half-life of ASOs in serum, explained by their improved nuclease-resistance and serum protein binding capacity. Some ASOs also include another modification in 2’ of the ribose molecule; such as 2ʹ¬O¬methyl (2ʹ¬OMe) and 2ʹ¬O-methoxy-ethyl (2ʹ¬MOE), which hybridize more efficiently to the target RNA and modify its expression. Another group is used to modify splicing or to inhibit translation. This group does not contain the typical deoxyribose in the phosphoribose backbone, but a morpho-lino ring instead. In addition, the linkage is an uncharged phosphorodiamidate one in-stead of a charged phosphodiester one. These properties make the ASOs even more re-sistant to nuclease and protease degradation. Depending on these properties, ASOs can modify the expression of a gene or the splicing of a pre-mRNA. The target region can be degraded by RNAse H (category I: RNAse H competent), or bind to block a start site, a RNA binding protein, a splicing site or an upstream open reading frame (uORF) (category II: steric block) [84]. Category I uses a RNAse H1 enzyme, which recognizes the DNA(ASO)-RNA complex and cleaves the binding site to degrade the matched RNA. Ap-proved ASOs from this category include fomivirsen, mipomersen and inotersen [85]. Cat-egory II uses a steric block which strongly binds to the target and masks this sequence during splicing or translation. Approved ASOs from this category include eteplirsen, go-lodirsen and nusinersen.
ASOs can be delivered using different methods such as (1) pairing the ASO to Trian-tennary N-acetylgalactosamine (GalNAc),(2) to peptides, (3) with lipids such as cholester-ol, (4) with antibodies or adaptamers, (5) or a stimuli-responsive structure. It is also possi-ble to pack the ASO in (6) a stable nucleic acid lipid particle or in an (7) exosome, (8) spherical nucleic acid nanoparticle made with a gold core linked to ASOs with met-al-thiol, (9) a DNA cage with an ASO at its end [85].
Clinical trials using ASOs show various efficacy. A Phase I/II clinical trial using AVI-4658, a morpholino ASO to skip the exon 51 from dystrophin in DMD patients (age 10 to 15). They received different doses of AVI-4658 through nine injections in the extensor digitorum brevis (EDB), a foot muscle. A biopsy of this muscle was made 28 days later to determinate if dystrophin, which was previously absent, could then be synthetized in the muscle. Immunohistochemistry and Western blots confirmed the presence of dystrophin in the fibres around injection sites [86]. Another Phase I/II trial with the same treatment injected intravenously in 5 to 15 year-old patients on a weekly basis for 12 weeks also showed up to 55% dystrophin-positive fibres for one patient, but the average [87]. For LGMDs, ASOs were proven effective to skip exon 32 in dysferlin in vitro, but clinical trials have not been done on humans yet [88].
Lines 225-242: In gene therapy descriptions, it is not clear whether the observations are made on mice experiments or clinical trials. I precised in the text now.
Line 249 (now 381): I also added that the treatment is on mice.
Line 272: I explained the abbreviation tracrRNA: In fact, the targeted sequence matches the complementary region RNA (crRNA) which further anneals to a trans-activating CRISPR RNA (tracrRNA). TracrRNA plays a role in the maturation of the CRISPR-RNA by pairing the preCRIPR-RNA, which will then be cleaved to become the guide for Cas9. Jennifer A. Doudna and Emmanuelle Charpentier fused both tracrRNA and crRNA together as a 18-24 nucleotides single guide RNA (sgRNA) also including a Cas9 domain to cleave a custom target sequence [103, 104].
